# Slowing deforestation in Indonesia follows declining oil palm expansion and lower oil prices

**David L. A. Gaveau** [1]*, **Bruno Locatelli** [2], **Mohammad A. Salim**[1], **Husnayaen**[1], **Timer Manurung**[3], **Adrià Descals**[4], **Arild Angelsen**[5], **Erik Meijaard**[6,7], **Douglas Sheil** [8,9]

**1** TheTreeMap, Martel, France, **2** CIRAD Forests and Societies, Univ Montpellier, Montpellier, France, **3** Auriga Nusantara, Bogor, Jawa Barat, Indonesia, **4** CREAF, Centre de Recerca Ecològica i Aplicacions Forestals, Bellaterra (Cerdanyola de Vallès), Catalonia, Spain, **5** School of Economics and Business, Norwegian University of Life Sciences (NMBU), Ås, Norway, **6** Borneo Futures, Spg 88, Kg Kiulap, Bandar Seri Begawan, Brunei Darussalam, **7** Durrell Institute of Conservation and Ecology, University of Kent, Canterbury, United Kingdom, **8** Department of Ecology and Natural Resource Management (INA), Norwegian University of Life Science (NMBU), Ås, Norway, **9** Forest Ecology and Forest Management Group, Wageningen University & Research, Wageningen, The Netherlands

* d.gaveau@thetreemap.com

**Data Availability Statement:** Data are available publlicly for interactive viewing at: https://nusantara-atlas.org/.

## Abstract

Much concern about tropical deforestation focuses on oil palm plantations, but their impacts remain poorly quantified. Using nation-wide interpretation of satellite imagery, and sample-based error calibration, we estimated the impact of large-scale (industrial) and smallholder oil palm plantations on natural old-growth ("primary") forests from 2001 to 2019 in Indonesia, the world's largest palm oil producer. Over nineteen years, the area mapped under oil palm doubled, reaching 16.24 Mha in 2019 (64% industrial; 36% smallholder), more than the official estimates of 14.72 Mha. The forest area declined by 11% (9.79 Mha), including 32% (3.09 Mha) ultimately converted into oil palm, and 29% (2.85 Mha) cleared and converted in the same year. Industrial plantations replaced more forest than detected smallholder plantings (2.13 Mha vs 0.72 Mha). New plantations peaked in 2009 and 2012 and declined thereafter. Expansion of industrial plantations and forest loss were correlated with palm oil prices. A price decline of 1% was associated with a 1.08% decrease in new industrial plantations and with a 0.68% decrease of forest loss. Deforestation fell below pre-2004 levels in 2017–2019 providing an opportunity to focus on sustainable management. As the price of palm oil has doubled since the start of the COVID-19 pandemic, effective regulation is key to minimising future forest conversion.

## Introduction

Concern for Indonesia's unique rain forests and their species-rich communities, including charismatic animals such as orangutans, tigers, and elephants is nothing new [1] but in recent decades this has increasingly focused on the palm oil industry [2]. This multi-billion dollar industry is based on cultivation of the African oil palm (*Elaeis guineensis* Jacq.) and conversion

**Funding:** This work was funded by the WWF-US and Global Environment Facility (GEF) under the Good Growth Partnership, and in collaboration with the Trase Initiative. The funders had no role in study design, data collection and analysis, decision to publish, or preparation of the manuscript.

**Competing interests:** The authors have declared that no competing interests exist.

to oil palm plantations has been highlighted as a major cause of deforestation and biodiversity loss [3]. In Indonesia, where half of global palm oil production occurs [4], expansion of oil palm plantations has often replaced forests. Indonesia is a focus of conservation efforts because it has some of the world's most remarkable and species rich forests and rapid deforestation [5].

A growing number of consumers demand palm oil-based products that are not associated with causing forest loss. The European Union has also become increasingly concerned about avoiding deforestation-tainted imports, especially for biofuel [6]. Other nations, together with environmental NGOs, have also sought to eliminate palm oil, or deforestation-linked palm oil, from consumer goods and other imports. Many of the world's largest traders and producers of palm oil have made "*No Deforestation*" commitments that guarantee to eliminate deforestation from their supply chain by a stated date [7]. Furthermore, in 2011, the Indonesian Government instituted a nationwide moratorium on developing new oil palm concessions on peatlands and in "primary forests", excluding natural forests impacted by selective timber harvesting, and reclassified as "secondary" forest by the Indonesian Ministry of Environment and Forestry [8]. The moratorium was extended indefinitely in 2019.

While many have blamed oil palm expansion for Indonesia's deforestation, this is contested by the industry, government and others [9]. One sample-based analysis estimated that between 2001 and 2016 industrial oil palm plantations (i.e. intensively managed large-scale, typically covering several thousand hectares of land, plantations owned by companies) accounted for 23% of total Indonesia-wide deforestation [10]. Taken over a longer period, 1972–2015, a regional study found that in Kalimantan (Indonesian Borneo), industrial oil palm accounted for just 15% of total deforestation because most new plantations made use of land cleared decades before [11]. However, from 2001 to 2017 conversion increased, accounting for 36% of total deforestation in Kalimantan [12]. These studies demonstrate that the impacts of oil palm on forests vary by region and period [13]. Several recent studies have demonstrated that industrial plantation expansion and associated forest conversion had slowed [10, 12] though not everyone agrees [14]. One reason for the slow-down might be, as indicated in a previous study from Kalimantan, that expansion of plantations is linked to the price of palm oil which has declined in recent years [12]. There are also concerns that even if expansion of industrial plantations slows this may be replaced by less readily observed growth among smallholders [15].

Those previous studies employed imagery from Landsat satellites to identify and measure changes in oil palm and forest cover over several decades. Due to limitations in detection, they omitted smallholder plantings. Industrial plantations are relatively easy to detect with satellite imagery because they exhibit distinctive linear boundaries, harvesting trails are laid out in grids on level land or follow contours on hilly terrain (Fig 1A and 1B). Smallholder plantings

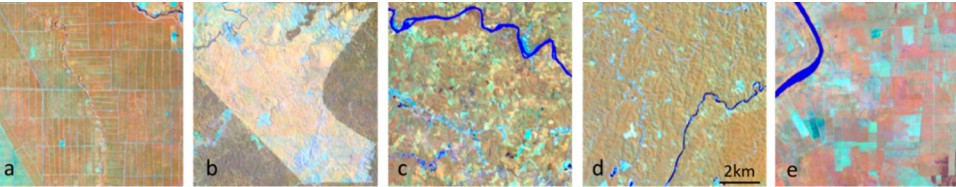

**Fig 1. Industrial and smallholder oil palm plantations seen by LANDSAT imagery (in 1:50,000 scale).** Imagery displayed in false colors (RGB: Near infrared; Short-wave infrared; Red). Here, closed canopy oil palm appears brown, open-canopy oil palm has different shades of yellow/orange. forest is dark brown. Recently cleared areas and newly planted areas appear bright cyan. (a) industrial plantations on a flat surface, with harvesting trails built in straight lines and thus forming rectilinear grids. (b) Young industrial plantation on hilly terrain with rectilinear borders. (c) smallholder plantations forming a mosaic with other types of landcover. (d) smallholder plantations joining together to form one large oil palm landscape. (e) Sometimes smallholder plantations owned by wealthy individuals extend several hundreds of hectares and resemble the linear structures of industrial plantations, although with less structure.

are harder to detect because they are typically smaller—< 25 hectares according to government definition—although wealthy individuals sometimes own several hundred hectares [16] —and their patterns are less consistent. Smallholder landscapes sometimes form a mixed mosaic with one or more other crops and types of landcover (Fig 1C), or a large homogeneous landscape (Fig 1D), or resemble industrial plantings though generally smaller and with less consistent structure (Fig 1E). The Indonesian government estimates that smallholder plantations constitute 40% of the total area under oil palm [17].

This article describes two decades of Indonesia's oil palm expansion and forest loss. For the first time, we present an annual time-series showing the expansion of industrial and smallholder plantations, forest loss, and their overlap, from 2001 to 2019. These data derive from complete annual maps created by interpretation of annual cloud-free Landsat composites, SPOT-6 and UAV imagery [18], combined with previously published sources of primary forest cover [5] and tree loss [19]. We separate results by regions to allow for different contexts (Sumatra, Kalimantan, Papua, Sulawesi, Java and Maluku). We compare our results against two previously published studies [10, 20] and against statistics from the Government of Indonesia [17] to determine similarities and differences. We also examine the links with annual prices of crude palm oil and discuss the implications.

## Methods

### Annual loss of forest

Annual forest loss represents the area of old-growth ("primary") natural forest that has been cleared each calendar year from 2001 until 2019. This measurement is based on the annual *Tree Loss* dataset (version 1.7) developed at University of Maryland with Landsat time-series imagery [19], which measures the removal of trees (height >5m) if the canopy cover of a 30 m x 30 m land unit (one Landsat pixel) falls below 30%. We corrected this dataset to improve consistency, i.e. we removed a number of omission and commission errors by scanning a sequence of annual cloud-free Landsat composites (see below), in particular for years preceding 2011 because Version 1.7 has an improved detection of tree loss since 2011, but the years preceding 2011 have not been reprocessed, so there were inconsistencies when we examine time-series over this period.

The *Tree Loss* dataset does not distinguish between different forest types. Thus, removal of old-growth forests (tree cover >80%) and regrowth or planted trees (where canopy cover can also be >80% and tree height >5m) can both appear as deforestation. Because our focus is on old-growth forests, we used a natural evergreen "primary" forest area mask for year 2000 developed previously [5], and excluded *Tree Loss* pixels outside of the area occupied by old-growth "primary" forests in year 2000, to determine losses in forest area rather than losses in planted trees. Old-growth forests usually have closed canopies (>80% cover) and high carbon stock (above ground carbon: 150 – 310 Mg C/Ha). They typically consist of tall evergreen dipterocarps growing on drylands or in swamps (including peat-swamps). There is considerable variation within and among all these forests. For example, on peat domes, forests may naturally be thinner, low carbon stock pole forests. In coastal regions, forests include mangroves as well as natural stands of Sago palm (*Metroxylon sagu* Rottb.). Here, "Forests" include intact and selectively harvested old-growth forests. Intact forests have either escaped significant recent cutting or modification by people, or such modifications were too minor to be detected. Selectively harvested forests have been subjected to industrial scale mechanized selective timber cutting and extraction but are recovering [21]. Intact and selectively logged forests are called "primary" and "secondary" forests on the Indonesian Ministry of Forestry and

Environment's forest maps [22]. Our definition of "forests" excludes young forest regrowth, agro-forests, mixed gardens, scrublands, tree plantations, agricultural land, and non-vegetated areas.

## Annual expansion of oil palm plantations

**Industrial oil palm.** To map the annual expansion of industrial plantations we scanned our sequence of annual cloud-free Landsat composites from 2000 to 2019. We looked at these images to confirm is they were cleared for oil palm based on characteristic features including: i) rapid changes in spectral colour indicating a transition from vegetation to bare land, ii) large cleared areas having boundaries with straight lines, and contour-like or grid patterns (Fig 1A and 1B and S1 Fig). We delineated the boundaries of plantations in 1:50,000 using a visual, expert-based interpretation method. We also employed maps of oil-palm concessions, reviewed online and press reports, and spoke to many experts to locate plantations. We defined an area "converted to industrial oil palm" the year an area that the characteristics of industrial oil palm plantations first appeared on our sequence of imagery. This identified areas cleared for oil palm. The land may or may not have subsequently been planted with oil palm trees (*Elaeis guineensis* Jacq.), and the trees may or may not have survived. We also mapped the expansion of industrial pulp and paper plantations (mainly *Acacia mangium* Willd., and *Acacia crassicarpa* A.Cunn ex Benth)), as these were clearly discernible from industrial oil palm (S2 Fig).

We determined the area of forest converted to industrial oil palm and pulp and paper plantations by measuring overlap between mapped plantations and mapped forest loss. We declared an area of forest "rapidly" converted, if there were already clear indications that this area was being developed as a plantation in the same calendar year that it lost forest cover. We reasoned that industrial plantations developed so rapidly after forest clearance are likely to be responsible for that clearance.

**Smallholder oil palm.** Mapping expansion of smallholder oil palm required a different procedure. We adapted an Indonesia-wide government-sanctioned "oil palm" base map developed by the NGO AURIGA in *circa* 2016–2018 [18]. This base map derived from a visual interpretation of high resolution (1.5 m) SPOT 6 satellite imagery acquired in 2016, complemented by aerial photography from a UAV (0.2–0.5 m resolution) taken in 2018 within key smallholder landscapes, and by Landsat imagery (15–30 m) [18]. We updated this map to 2019 using radar data [20], and estimated the year of expansion from 2001 to 2019 using the annual *Tree Loss* dataset described above [19]. We outline those steps below.

First, we merged the AURIGA base map of oil palm with our own map of 2019 industrial oil palm extent (developed using Landsat, see paragraph above). Those areas classified as "oil palm" on the AURIGA base map but not in our Landsat evaluations were considered as "smallholder oil palm". Second, we updated the AURIGA smallholder oil palm base map to 2019 by using an existing classification of a 2019 radar composite [20] (S3 Fig). We note, however that this 2019 update may be incomplete because radar imagery can fail to detect young plantations (< 3 years) [20]. We examined and quantified this underestimation through a rigorous validation (see below). Third, we removed coconut plantations misclassified as oil palm in the coastal southeastern district of Indragili Hilir, Riau Province, (Indonesia's largest coconut plantations) using a map from the provincial government (S4 Fig). Fourth, we estimated the year smallholder plantations were established using the annual *Tree Loss* dataset described above. This method assumes that all expansion from 2001–2019 remain bounded by the existing 2019 smallholder map, which is reasonable because oil palm plantations are established for the long term (25 years), therefore unlikely to be subsequently converted to another land use.

It also assumes that year tree loss was indicated is also the year the smallholder oil palm was initiated—we already know that 92% of all forest conversion to industrial plantations from 2000 to 2017 in Borneo occurred within one year of clearance [12]. Areas detected as "oil palm" by the AURIGA base map that did not experience any tree loss during the 2001–2019 period were classified as smallholder oil palm plantations that already existed in 2000. This approach cannot account for smallholder plantations established after 2000 on open land that lacked tree cover before 2001, thus the total expansion of smallholder plantations may be underestimated. However, this potential underestimation, does not affect our main goal here which concerns assessment and attribution of forest conversion.

## Map validation

We validated the 2019 oil palm map by visually detecting the presence or absence of oil palm at 3,440 randomly sites (one site = one pixel of 30 m x 30 m) using several image sources. We selected the sites using a stratified-random sampling approach in which we first selected (i) 635 sites from the areas classified 'industrial oil palm', (ii) 398 sites in areas classified as 'smallholder', and 2,407 sites classified as 'Other'. We used these reference data to calculate the overall accuracy (OA), producer's accuracy (PA), and user's accuracy (UA) with a 95% confidence interval of the 2019 'industrial' and 'smallholder' oil palm maps following "good practices" for estimating area and assessing accuracy [23]. We also calculated Cohen's kappa coefficient [24].

We employed all the free high-resolution imagery (<1 m) available in Google Earth Engine and in ESRI's World imagery service as the highest level of proof to describe the 3,340 reference sites as either 'Industrial oil palm', 'Smallholder oil palm', or 'Other' because these images detect individual oil palm tree stands (S5 Fig). These images were taken at different dates, however, and for those sites where the high-resolution imagery was older than 2019 and showed no oil palm tree stands, we filled the gap with a 2019 radar Sentinel-1 composite (10 m) developed previously [20], and a 2019 cloud-free optical Sentinel-2 composite (10 m). Radar reveals a distinct backscatter for closed canopy oil palm stands (> 3 years), so was used to further check the presence of closed-canopy oil palm stands in areas where plantations were established after the high-resolution imagery was taken. We employed the 2019 cloud-free Sentinel-2 composite (10 m) to indirectly identify young plantations, over reference sites where the high-resolution imagery was too old, and the 2019 radar backscatter inconclusive.

If we detected oil palm stands over a reference site on the high-resolution imagery, we labelled this site 'Oil palm'. If the site was inside a well delineated plot of land with oil palm tree stands, but without oil palm tree directly in the site, this was labelled 'Oil palm' too, indicating a damaged plantation (S5 Fig). If we detected these stands within landscapes where plantation boundaries formed straight lines, and harvesting trails formed rectilinear or contour grid, the reference site was classified 'Industrial oil palm' (S5 Fig). If we detected the stands in a landscape mosaic with no clear straight boundary, no rectilinear trails, the reference site was labelled 'Smallholder oil palm'. We distinguished coconut and oil palm stands based on the larger spacing between trees and more irregular planting patterns of coconut plantations. We also employed a provincial map of coconut plantations from Ministry of Agriculture (S4 Fig). Given the subjective nature of these choices some uncertainty remains, especially in the smallholder data in locations where both crops co-exist.

For reference sites, where high-resolution imagery was either not available or did not indicate any oil palm stands, and where the 2019 radar imagery did not produce the distinct backscatter of oil palm stands, we employed the 2019 cloud-free optical Sentinel-2 composite (10 m) to infer the presence or absence of young oil palm over the site. A plantation may have been established after the high-resolution imagery was taken and may be too young to be

detected by the 2019 radar composite. If we observed no change in land cover between the high-resolution imagery (devoid of oil palm) and the 2019 sentinel-2 composite, this site was labelled 'Other' (S6 Fig). If we observed a clearing typical of industrial plantations (rectilinear grids, straight boundaries) appear in the 2019 Sentinel 2 composite that was not on the high-resolution imagery, we labelled this site as 'Industrial oil palm' (S7 Fig). Concession information was also used to separate clearing for oil palm from clearing for pulp wood because these can exhibit similar planting patterns. If we observed a clearing typical of smallholder plantations on the 2019 imagery, we labelled the site 'smallholder oil palm' if this clearing was adjacent to existing smallholder oil palm plantations seen on the high-resolution imagery (S7 Fig). If the landscape did not indicate any oil palm in the neighbourhood, the cleared site was labelled 'unknown', and were ultimately excluded (77 sites) from the reference dataset. The final sample size explored here, N = 3,440, excludes these discarded points.

We employ the term '*mapped oil palm*' for the area classified as oil palm on the map. We employ the term '*adjusted oil palm*' for the estimation of the oil palm plantation extent based on the validation of our map against the reference dataset. For instance, a high omission rate in the 'smallholder' class would potentially lead to a lower *mapped area* than an *adjusted area* for that estimate, while a high commission rate would potentially lead to a higher *mapped area* than the *adjusted area*. The *adjusted area* represents an estimation of the actual oil palm extent for year 2019. The accuracy of the map, and the sample size of the reference dataset, play a role in the confidence interval of *adjusted area* estimate. Lower map accuracy and smaller sample size mean wider confidence intervals.

**Time-series validation.**　We visually interpretated original Landsat images acquired between January 2000 and December 2019 over the reference sites labelled 'Industrial oil palm' (N = 612) to verify the year industrial plantations were established. The visually interpreted Landsat images were obtained from the Landsat-5, Landsat-7, and Landst-8 surface reflectance datasets, that have a combined revisit time of 8 days. For each reference site, we plotted the values of the normalised burned ratio (NBR), an index ranging from -1 to +1, and looked for a sudden drop in the NBR time series, indicating a disturbance in the vegetation [25], possibly a conversion to oil palm (S8 Fig). We reviewed the images before and after the drop in NBR to verify whether the disturbance was a clearing event to establish oil palm. If the border of the cleared area was linear or rectangular, and grid or contour-like trails appeared on the sequence of Landsat images, this indicated an area converted to an industrial oil palm plantation, and we recorded the year of conversion (S8 Fig). Finally, we analysed the correspondence between the year of conversion observed by visual interpretation and the year of conversion of our oil palm map. To assess the replicability of our data, we also compared our annual forest-to-industrial-oil palm conversion trends against those reported in a previous study from 2001 to 2016 [10]. The study used a sample-based approach to estimate the area of forest converted to industrial oil palm by reviewing Landsat and high resolution imagery over a sample of points seen cleared by the same *Tree Loss* loss dataset used here [19]. We could not replicate these validations for smallholders because the spatial patterns of smallholder oil palm are not immediately obvious with Landsat and the large volume of high-resolution imagery (< 1 m) required every year for the period considered is unavailable.

## Developing annual landsat composites

We generated annual cloud-free Indonesia-wide Landsat composites for each year from 2000 to 2019 with the Google Earth Engine [26]. The cloud observations in the Landsat images were firstly masked with the quality band 'pixel_qa', which is generated from the CFMASK algorithm and included in the Surface Reflectance products [27]. Then, for each year, we created

the annual composites using two criterions: 1) the median pixel-wise value of the Red, Near Infrared (NIR), and Shortwave Infrared (SWIR) bands of the images acquired between 1 January and 31 December, and 2) the minimum pixel-wise normalized burned ratio (NBR) of the images taken in the same given year. The composite image based on the median produces clean cloud-free mosaics but tends to omit new plantations developed at the end of the year. The second approach, based on the minimum NBR, produces noisier composites (residual clouds and shadows may persist), but it presents the advantage to capture plantations developed at the end of the year.

## Results

### Summary 2001–2019

We estimated that 87.76 Mha, or 46% of Indonesia's land, was natural forest in 2019. Lowland forests (<500 m asl), among the world's most species-rich [28] covered 55.72 Mha. By comparison, the Indonesian government reported 88.8 Mha of natural forest in 2017 [22]. The area of forest declined by 9.79 Mha (11%) from 2001 to 2019, representing an average annual loss of 0.51 Mha yr$^{-1}$. Sumatra and Kalimantan lost more forest than other regions with 4.08 Mha (25%) and 4.02 Mha (14%), respectively. A quarter (12.06 Mha) of Sumatra, and nearly half (25.74 Mha) of Kalimantan were forest in 2019. Papua (Indonesian New Guinea) lost the least (0.75 Mha; 2%) and retained the largest area (34.29 Mha, 83% of its landmass, or 41% of Indonesia's remaining forests). Sulawesi experienced similar losses to Papua (0.72 Mha; 7%) but retained less forest (9.11 Mha; 49% of its landmass). See Table 1 for a breakdown by regions.

From 2001 to 2019 Indonesia gained 8.48 Mha of oil palm plantations (6.19 Mha industrial; 2.28 Mha smallholder) reaching a total mapped plantation area of 16.24 Mha in 2019, with 64% industrial and 36% smallholder. The total mapped area developed as industrial

**Table 1. Share of deforestation caused by oil palm expansion from 2001 to 2019 for Indonesia and by region.**

| Areas (in Ha) | Indonesia | Sumatra | Kalimantan | Papua | Sulawesi | Java Maluku |
|---|---|---|---|---|---|---|
| **Landmass** | 189,130,128 | 47,467,842 | 53,498,290 | 41,227,232 | 18,627,593 | 21,135,660 |
| **2019 Forest area** | 87,758,114 (**46%**) | 12,063,230 (**25%**) | 25,742,162 (**48%**) | 34,289,462 (**83%**) | 9,114,005 (**49%**) | 5,871,624 (**28%**)* |
| **2019 Forest area (<500 m asl)** | 55,724,906 | 5,961,949 | 17,266,990 | 25,165,882 | 3,130,233 | 3,920,071 |
| **Forest loss 2001–2019** | 9,789,448 (*11%*) | 4,075,312 (*25%*) | 4,023,971 (*14%*) | 748,640 (*2%*) | 715,737 (*7%*) | 213,487 (*4%*) |
| **Forest converted to OP** ‡ | 3,094,882 (32%) | 1,242,345 (31%) | 1,593,260 (40%) | 200,161 (27%) | 46,782 (7%) | 12,629 (6%) |
| **Rapid conversion**§ | 2,849,796 (29%) | 1,166,806 (29%) | 1,434,493 (36%) | 194,996 (26%) | 43,319 (6%) | 10,181 (5%) |
| **Rapid conversion to industrial OP** | 2,129,301 (22%) | 553,480 (14%) | 1,341,610 (33%) | 194,671 (26%) | 29,807 (4%) | 9,733 (4.5%) |
| **Rapid conversion to smallholder OP** | 720,495 (7%) | 613,326 (15%) | 92,884 (3%) | 325 (0.0004%) | 13,512 (2%) | 448 (0.002%) |

We used a sinusoidal projection to calculate areas.

OP: Oil Palm

(**%**) of landmass.

(*%*) of 2000 forest area.

(%) of forest loss.

‡ Area of forest in 2000 and converted to oil palm plantation by 2019. N.B. we cannot assert that all 3.09 Mha were cleared for oil palm as they may have been cleared for other reasons before subsequently being planted with oil palm.

§ The area of forest that was replaced by oil palm in the same year it was cleared. We can assert that all 2.85 Mha were cleared by oil palm expansion.

* Maluku lost 201,081 ha of forest between 2001 to 2019. It had 5,167,788 ha forest left in 2019, or 66% of its landmass (7,876,562 ha). Java lost 12,406 ha. It had 703,836 ha forest left, or 5% of its landmass (13,259,098 ha).

The provinces of Bali and East and West Nusa Tenggara (landmass = 7,173,511 ha) lost 12,301 ha of forest between 2001 and 2019, representing 2% loss. In 2019, there were 677,631 ha of forest and no oil palm in these three provinces. These regions have no oil palm.

plantations reached 10.32 Mha (64%) in 2019 (minimum size of a plantation = 84 ha; max = 185,000 ha; mean = 3,000 ha). Smallholder plantations reached 5.92 Mha (36%) in 2019 (minimum size of a plantation = 0.29 ha; max = 149,000 ha; mean = 49 ha). We note that the maximum size for smallholders corresponds to a large homogenous landscape made up of several small plantations.

In this assessment, smallholder plantations developed in plasma schemes [16] were counted as "industrial" because the patterns look like industrial plantations in satellite images. We note that our total mapped estimate includes immature, damaged, and failed plantations and thus surpasses estimates that include only mature and relatively intact closed-canopy plantations [20] (e.g., 11.5 Mha). The Government of Indonesia's estimates based on company reports and interview with smallholders (14.72 Mha), include mature and damaged plantations [17], but likely omit some illegal plantations that go unreported. We estimate that oil palm plantations in the State Forest Zone (*Kawasan Hutan*), where oil palm is prohibited, covered 3.13 Mha in 2019, i.e., 19% of total oil palm area. See Table 2 for a comparison of datasets, and breakdown by regions considered.

The overall accuracy of our 2019 mapped plantation extent is 95.6% (CI: 95.3%-96.0%) and Cohen's Kappa coefficient is 0.85. We report user's accuracies (UA) for the 'industrial' and 'smallholder' classes, at 95.3% (CI: 94.5%-96.2%) and 87.4% (CI: 85.8%-89.1%) respectively, indicating 4.7% and 12.6% commission-error rates (Table 3 and S1 and S2 Tables). The producer's accuracies (PA) are comparatively lower, but notably less so for the 'industrial' class, at 91.7% (CI: 90.3%-93.2%) than for the 'smallholder' class at 63.9% (CI: 61.4%-66.4%), indicating an omission error of 8.3% for industrial plantations, and an omission error of 36.1% for smallholders. Therefore, while our map indicates 16.24 Mha of oil palm in 2019, with 64% (10.31 Mha) industrial and 36% (5.92 Mha) smallholder, the omission-adjusted estimate for the total area under oil palm is 18.83 Mha (CI: 18.3–19.4), with 10.72 Mha (57%) being industrial and 8.11 Mha (43%) being smallholder.

The mapped area that was forest in 2000 and oil palm in 2019 is 3.09 Mha (32% of total forest loss: 9.79 Mha), with 2.85 Mha (29%) cleared and converted in the same year (termed "rapid conversion" in Table 1): 2.13 Mha (22%) by industry and 0.72 Mha (7%) by smallholders. In general, more plantations were established in areas cleared of forest before 2000 (5.39

**Table 2. Oil palm expansion from 2001 to 2019 and planted area in 2019 for Indonesia and by region.** Based on three different sources: this study, a global study and government statistics.

| Areas (in Ha) | Indonesia | Sumatra | Kalimantan | Papua | Sulawesi | Java Maluku |
|---|---|---|---|---|---|---|
| **Oil palm expansion 2001–2019** | 8,477,253 | 3,457,500 | 4,598,415 | 221,117 | 164,471 | 35,749 |
| **Oil palm area 2019 (This study)** | 16.237,047 | 9.486,516 | 6.044,517 | 272,808 | 374,686 | 58,520 |
| **Industrial** | 10,316,986 (64%) | 4,684,385 (49%) | 5,105,427 (84%) | 271,486 (99.5%) | 207,165 (55%) | 48,522 (83%) |
| **Smallholder** | 5,920,061 (36%) | 4,802,130 (51%) | 939,091 (16%) | 1,322 (0.5%) | 167,520 (45%) | 9,998 (17%) |
| **Oil palm area 2019 (Descals et al. 2020)*** [20] | 11,531,006 | 6,770,223 | 4,259,152 | 175,803 | 304,442 | 36,379 |
| **Industrial** | 7,706,254 (67%) | 3,692,628 (55%) | 3,682,299 (86%) | 169,880 (97%) | 144,787 (48%) | 27,556 (76%) |
| **Smallholder** | 3,828,849 (33%) | 3,077,595 (45%) | 576,853 (14%) | 5,923 (3%) | 159,655 (52%) | 8,823 (24%) |
| **Oil palm area 2019 (Ministry of Agriculture 2020)‡** [17] | 14,724,420 | 8,299,729 | 5,713,504 | 213,359 | 450,499 | 47,328 |
| **Industrial** | 8,688,678 (59%) | 3,560,687 (43%) | 4,670,281 (82%) | 180,685 (85%) | 238,498 (53%) | 38,527 (81%) |
| **Smallholder** | 6,035,742 (41%) | 4,739,042 (57%) | 1,043,223 (18%) | 32,674 (15%) | 212,001 (47%) | 8,801 (19%) |

*Area of plantations extracted from a global oil palm map derived by based on radar data [20]. This dataset only includes mature (closed-canopy) plantations

‡Area of plantation extracted from 2019 statistics of the Directorate General of Plantation Estates Crops of the Indonesian Ministry of Agriculture [17]. This dataset includes immature (open-canopy), mature (closed-canopy) and damaged plantations.

**Table 3. Accuracy assessment of the Indonesia-wide 2019 oil palm plantation extent.** The accuracy metrics were estimated with an initial total of 3,340 reference sites randomly distributed using stratified sampling in non-forest areas and below 500 m asl. The reported metrics are: 1) the overall accuracy (OA), the user's accuracy (UA), and the producer's accuracy (PA) with their 95% confidence intervals, and 2) the mapped oil palm extent (industrial and smallholder) and the adjusted extent with their 95% confidence intervals.

| | | |
|---|---|---|
| OA (%) | | 95.6 (95.3, 96.0) |
| | Industrial oil palm | 95.3 (94.5, 96.2) |
| UA (%) | Smallholder oil palm | 87.4 (85.8, 89.1) |
| | Other | 96.1 (95.2, 96.0) |
| | Industrial oil palm | 91.7 (90.3, 93.2) |
| PA (%) | Smallholder oil palm | 63.9 (61.4, 66.4) |
| | Other | 99.0 (98.9, 99.1) |
| Mapped Industrial oil palm (Mha) | | 10.31 |
| Adjusted Industrial oil palm (Mha) | | 10.72 (10.53, 10.92) |
| Difference (Mha) | | 0.41 |
| Mapped Smallholder oil palm (Mha) | | 5.92 |
| Adjusted Smallholder oil palm (Mha) | | 8.11 (7.77, 8.44) |
| Difference (Mha) | | 2.19 |
| Percent Ratio (Industrial/smallholder): | | |
| Mapped | | 64% / 36% |
| Adjusted | | 57% / 43% |

Mha; black and white bars in Fig 1). Only in Papua did most new plantations replaced forests (Fig 2).

Comparing regions, we find that Kalimantan experienced the highest share of both total and rapid (same year) forest conversion to oil palm (40% and 36% respectively), followed by Sumatra (31% and 29%), Papua (27% and 26%) and Sulawesi (7% and 6%). In Sumatra and Sulawesi, industrial and smallholder driven conversion were similar in magnitude while industrial conversion dominated in Kalimantan and Papua (Table 1).

## Annual trends

Industrial and smallholder plantations followed similar trends: expansion (black and white bars; Fig 1A–1C) and rapid forest conversion to oil palm (white bars only) increased during the 2000s, peaked in 2009 (0.84Mha added; 0.28 Mha forest converted) and 2012 (0.80 Mha added; 0.31 Mha forest converted) and steadily declined thereafter. In 2019, overall plantation expansion had dropped to pre-2004 levels (0.16 Mha added; 0.059 Mha forest converted). Verification of establishment year by visual interpretation of original Landsat images for the reference sites labelled 'Industrial oil palm' (N = 612) agreed well with the map: for 83% (N = 509) of sites, the verified year matched the year on the map, and in 13% of cases, the difference was only one year (S9 Fig). The difference between map and reference year of plantations was greater than 1 year only for 4% of sites. The year with the highest error was found for 2000. This error, however, only represents the 2.9% of all points, which indicates that the satellite data used in the study could observe the year of establishment in most of the pixels.

The trends for forests converted to industrial plantations annually (white bars in Fig 1B) also follow those from a previously published sample-based approach which estimated how much industrial plantations expanded into forests each year between 2001 and 2016. We report 1.97 Mha rapid forest conversion to industrial oil palm (sum of white bars Fig 1A) for the 2001–2016 period, while the previous study reported 2.08 Mha. Both datasets show a decline in industrial oil palm driven deforestation since 2011–12 (S10 Fig). The positive

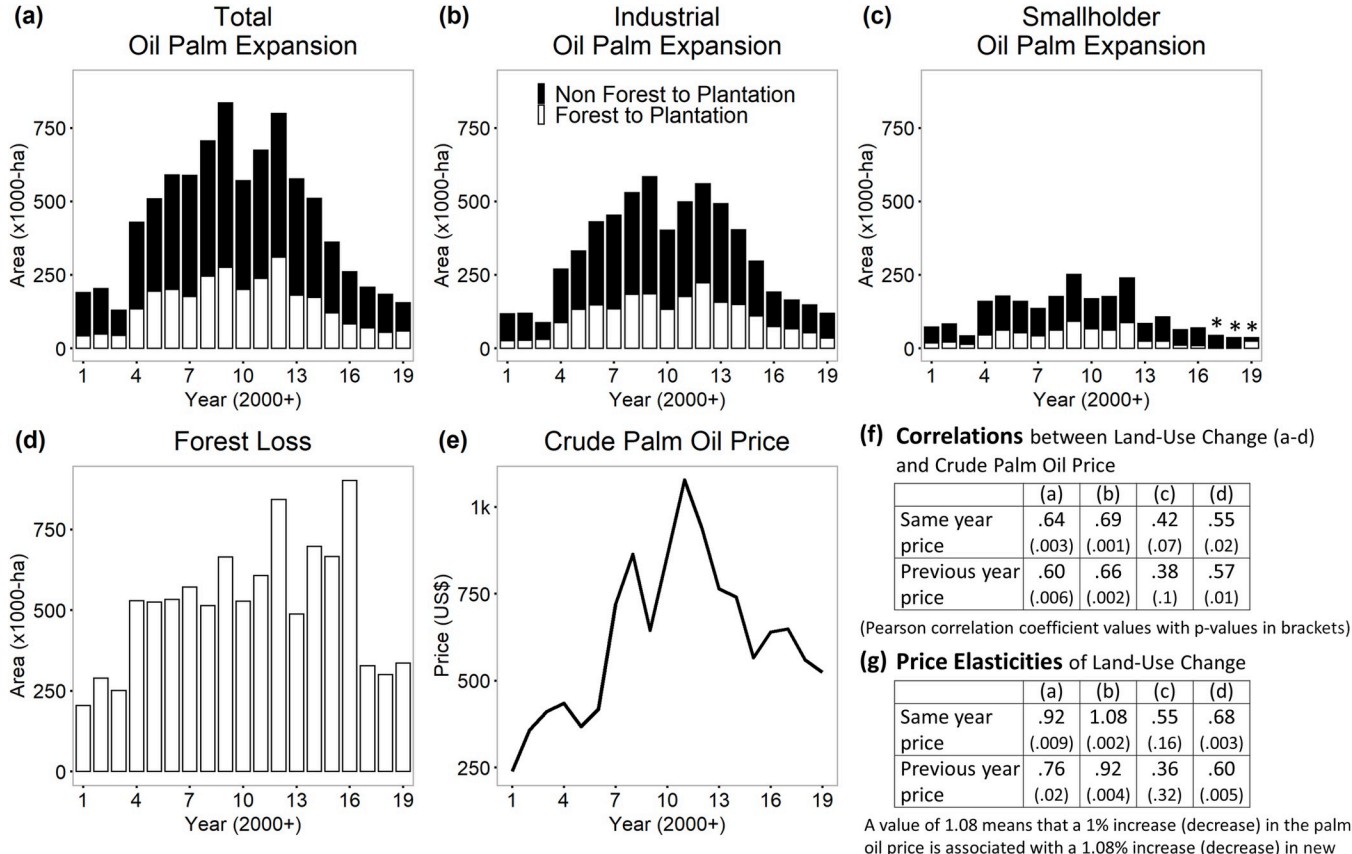

**Fig 2. Time-series of Indonesia's land-cover/use change from 2001 to 2019.** Expansion of oil palm plantations by year (a), split between industrial and smallholder (b,c). Forest loss (d). Mean annual Crude Palm Oil price (e). Correlations/elasticities with the previously shown land-cover change estimates (f,g). Price calculated from monthly prices in USD using IMF data [29]. In insets (a,b,c) white bars represent the areas of forest cleared and converted to plantations in the same year. This rapid conversion constitutes 29% of all forest loss detected during 2001–2019. The black bars represent areas of non-forest converted to oil palm. Ninety six percent (5.39 Mha) of those non-forest areas were non-forest in 2000, 4% (0.24 Mha) were forest cleared after 2000 and converted to plantations more slowly (after 2 to 18 years). These non-forest areas include conversion of young regrowth, mixed gardens, agroforests, and rubber plantations. Asteriks (*) indicate that the area of smallholder expansion in 2017, 2018, 2019 is likely underestimated. We note that the 2016 peak in forest loss (d) includes losses of late 2015, when fire burned large areas of forest in Kalimantan. Much of these losses were recorded only the following year by the *Tree Loss* dataset used to calculate forest loss because of cloud cover.

correlation between the estimations of the two studies is 90% (Pearson's correlation coefficient $r = 0.90$, p<0.001). We estimated that the annual mean area converted from forest to industrial oil palm is 123,000 ha (standard deviation 60,000) while Austin et al. 2019 [10] estimated a slightly higher 130,000 ha (standard deviation 72,000). A linear model with zero intercept of the former estimation against the latter with each shared year being one datum has a slope of 0.9 (0.90, p<0.01).

The expansion of new oil palm plantings has slowed in all regions (Fig 3 and S11 and S12 Figs). This occurred later in Papua where expansion peaked in 2015 rather than in 2012 as observed elsewhere. The market price of crude palm oil (CPO) has also risen and then declined over the period of our study with peaks in 2008 and 2011 (Fig 2E). We note a positive correlation between annual CPO prices [29] and expansion of oil palm plantations (also industrial plantations, but not significant for smallholders) as well as between CPO prices and forest loss (Fig 2F). A decrease/increase in CPO prices by 1% was associated with a decrease/increase of new industrial plantations by 1.08% and with a 0.68% decrease/increase of forest loss (Fig 2G).

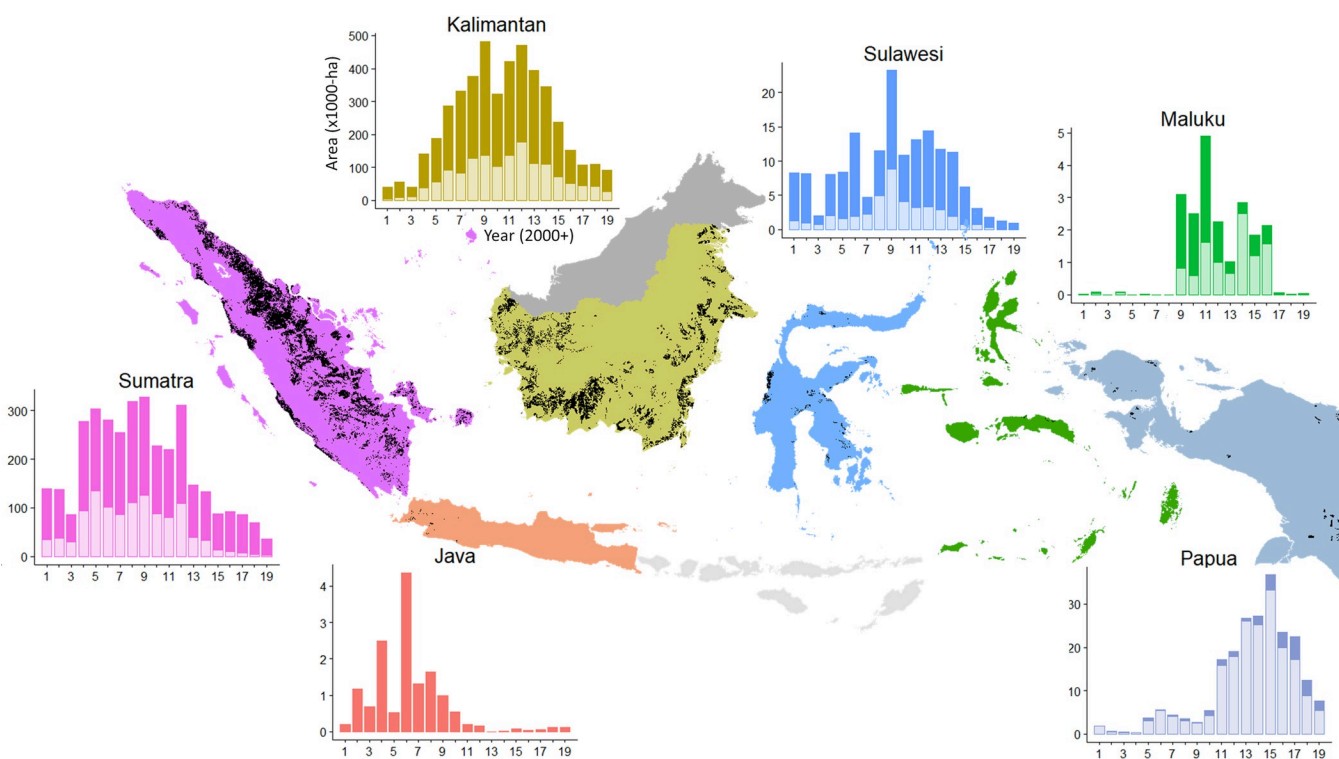

**Fig 3. Oil palm expansion from 2001 to 2019 by Indonesian region.** Y-axis represent areas (in 1000-ha, note different scales) of the total area of plantations (industrial and smallholder) added each year by rapidly clearing forests (light bars), or by using areas already cleared (dark bars). Black areas on the map represent the total 2019 oil palm area (industrial and smallholder combined).

Annual forest loss climbed in 2004, followed by variable but high rates that reached a maximum in 2016 (0.90 Mha cleared) before falling to pre-2004 levels (<0.34 Mha cleared) for three consecutive years (Fig 2D). This 2016 maximum is evident in Kalimantan and Sulawesi, but not in Sumatra where forest loss peaked in 2012 and in Papua where it peaked in 2015 (Fig 4). We note that the 2016 peak in forest loss includes large losses of late 2015, when fire burned large areas of forest in Kalimantan. Much of these losses were recorded only the following year by the *Tree Loss* dataset used to calculate forest loss because of cloud cover.

## Discussion

We estimated the annual expansion of industrial and smallholder oil palm plantations and their overlap with forest loss from 2001 to 2019 across Indonesia. Industrial plantations expanded faster than smallholder plantations (6.19 Mha added vs 2.28 Mha) and caused almost three times as much forest conversion (2.13 Mha vs 0.72 Mha). We find an increase in plantations expansion and associated forest conversion during the 2000s, followed by a decline after 2012 (Fig 2A–2C). We note that our annual values for forest conversion to industrial plantations, for the shared periods (2001–2016) closely follow those of Austin et al. 2019 [10] (S10 Fig), indicating the robustness of these patterns. We were more concerned to note that Xu et al. (2020) [14] reported a peak in expansion in 2016 but subsequent discussions with those authors suggested an artefact due to using multiple data sources with distinct and sometimes inconsistent properties. The authors recommend users excluding the last year map (2016) for further analysis [30]. We also observed an abrupt decrease of deforestation in 2017, 2018 and 2019. Our data show that expansion of plantations directly replaced 29%-32% of the total forest

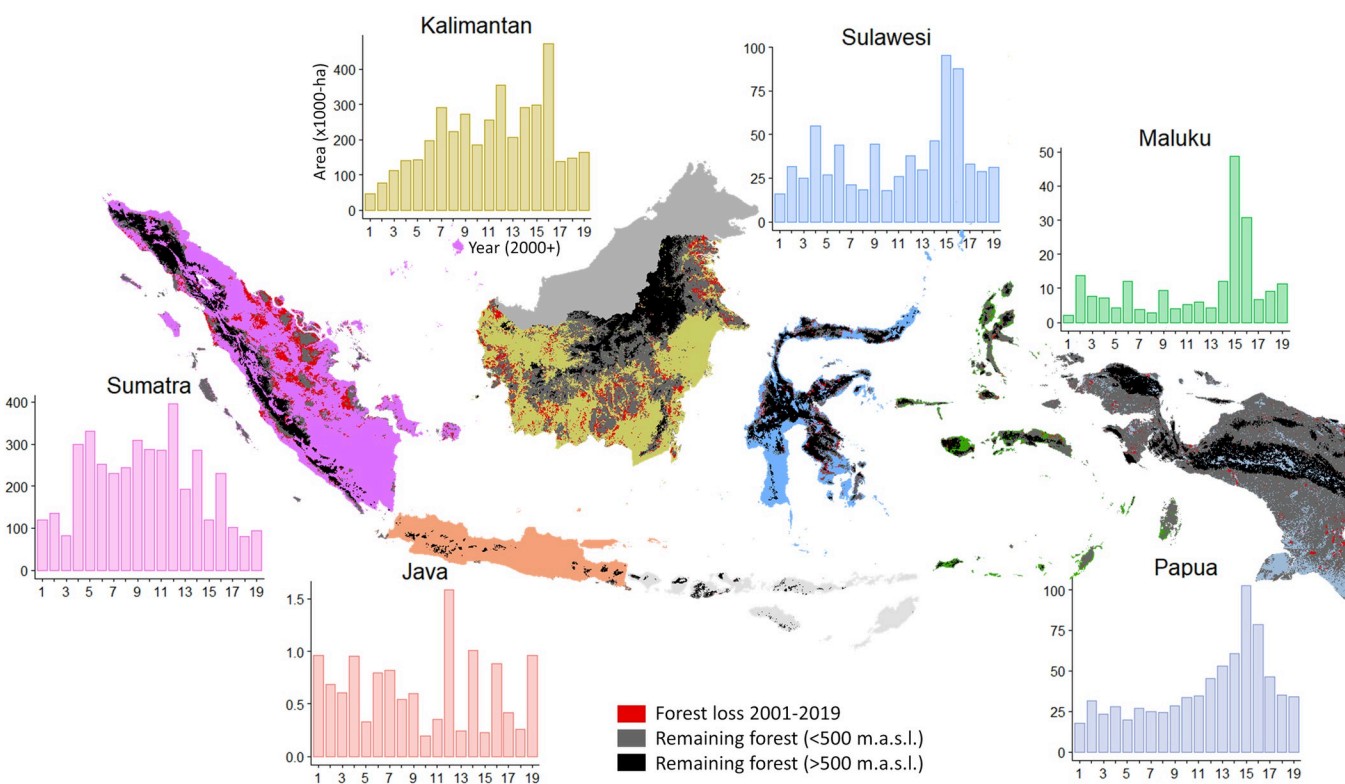

**Fig 4. Deforestation from 2001 to 2019 and remaining forest in 2019 by Indonesian region.** Y-axis represent areas (in 1000-ha, note different scales) of forest cleared each year.

area lost between 2001 and 2019. We conclude that oil palm was responsible for one-third of Indonesia's loss of old-growth forests over the last two decades (N.b. this neglects any impacts from associated infrastructure, immigration, and delayed conversion, see below). If we include impacts of industrial pulp and paper plantations, the direct conversion due to expansion of plantations directly accounts for 39%-42% of forests lost between 2001 and 2019 (S3 Table).

We underline that these estimates are conservative. Our Indonesia-wide 2019 oil palm map extent underestimated the true plantation extent, particularly for smallholder plantings (Table 3). The identification and quantification of less-readily-detected young (< 3 years) smallholder oil palm plantations, and scattered mixed plantings, remains a challenge that will require dedicated and targeted research. Given such uncertainties, we may not detect every change in planting rates, such as a small increase in smallholder oil palm planting after 2016. Our analysis also omits indirect effects on forest through road expansion, infrastructure and resulting in-migration. It also excludes conversion of young forest re-growth, agroforests, and other mixed gardens. For evaluations of impacts of oil palm on regrowth forests more complex remote sensing assessments and field measurements will be necessary. Nevertheless, our results confirm oil palm's major role as a direct driver of old-growth forest loss, but also reveal that recent expansion and associated forest conversion have declined. Our data also shows that industrial plantations caused more deforestation than smallholders. In addition, and contrary to expectations [15], detected smallholder plantations followed a similar decline as industrial plantations, suggesting that similar factors were at play. Such parallel trends may obscure subtle differences as the annual proportion of total national planted area due to smallholder oil palm is predicted to grow from 40% (we find 36% to 43%, Table 3) to over 60% by 2030 [15].

Past warnings regarding the fate of Indonesia's lowland forests appear justified [1, 28]. In Western Indonesia, particularly over much of Sumatra and even Kalimantan, remaining lowland forests are fragmented and scarce (see Fig 4). These forests have high conservation value but remain vulnerable to multiple threats including fire, conversion to plantations, hydropower dams and road developments [31–34]. Many also occur in a matrix of degraded forest and regrowth that has a potentially higher conservation value than is often acknowledged. They will recover even greater value if protected, and should be included in conservation planning [35]. In Western Indonesia strong conservation action is needed to manage and restore connectivity among forest areas. In contrast to Western Indonesia, Eastern Indonesia, and Maluku and Papua in particular, retain tracts of relatively pristine forests offering development trajectories that should include forest protection as a core value. Given the profitability of the crop the oversight of further oil palm plantings is likely to remain contentious, though efforts to increase transparency over sourcing have increased.

Why did plantation expansion slow after 2012? The positive correlation with Crude Palm Oil (CPO) prices suggests that finance was influential. The economies of China and India flourished from 2000 to 2011 and the price of CPO quadrupled (annual mean from USD 261 to USD 1077). This accelerated investment in new plantings. Around 2011–12, as the economies of China and India slowed, the market could not readily absorb the increased supply and prices declined. Subsequently, between 2011 and 2019, the annual price of CPO halved (USD 523 by 2019). Another reason for this price decline may be the influence of the crude oil price (i.e., "mineral oil") on the palm oil price [36], with the former reducing by nearly half from 2012 to 2019 [37]. Palm oil planting remains a risky investment due to the upfront expenses and the time required to recoup costs. New investments became less attractive when the Indonesian government introduced the deforestation moratoria in 2011 (extended in 2019) and additional attention focused on regulations and good practice [8].

Why has forest loss peaked in 2016 and slowed from 2017 to 2019? One factor is the similar trends in plantations already described. Another is fire. The 2016 peak (Fig 2D) reflects the El Niño-induced drought of late 2015, when fire burned large areas of forest in Central Kalimantan [38]. Much of these losses were recorded only the following year by the dataset we used to calculate forest loss (see Methods). One study estimated that forest conversion to grasslands due to fire explained 20% of total forest loss in Indonesia between 2001 and 2016 [10], though we note that such complete conversion typically results from a series of fires over several years, not from a single event as recorded by these authors. 2017 and 2018 were generally wetter resulting in less flammable landscapes. In contrast, 2019 was again a dry fire-prone year, though less forest was burned than in 2015 [39].

Might Indonesia have already reached the slowdown in forest loss expected for a forest transition? Probably not: despite the positive trends the rise-and-fall pattern we observe in Indonesia does not follow the forest scarcity and economic development that determine a genuine forest transition in other regions [40]. Forest remains abundant (46% of Indonesia's landmass) and any marked impact of economic development in slowing forest loss would not emerge over such a short period [41].

Various other factors have likely influenced the declining trends. For example, in many regions remaining forests are inaccessible or protected, and the improved legal basis for community-based land claims have likely further curtailed companies' access to land [42]. Furthermore, an increasing number of consumers seek products that they consider ethical. This has generated various initiatives aimed at distinguishing and certifying products that avoid deforestation, and companies seeking to be associated with pledging to avoid forest clearing with "no- deforestation commitments" [13]. This interest has led to wide scrutiny of the palm oil industry using data that now include publicly available satellite imagery in near-real time [43].

Taken over a longer period, we note that deforestation from 2001 to 2019, after Indonesia became a democracy, was only a third of that during the preceding two decades under the Suharto regime in 1980s and 1990s (0.51 Mha against 1.67 Mha) [1]. It was forest lost in such earlier times that has permitted many plantations to be developed without the need to replace old growth forests.

While slowing deforestation and commitments from industrial commodity producers to avoid forest loss justify cautious optimism, forest loss has not ceased and there is no guarantee that the low levels of conversion seen in 2017–2019 will remain. CPO prices are rising again (annual mean from USD 524 in 2019, to USD 666 in 2020 to USD 1039 in 2021), increasing the profitability of palm oil again, potentially driving further expansion. Additional demand, driven by Indonesia's biodiesel program, could stimulate expansion. The forests excluded from the Moratorium would be at particular risk (i.e., old growth forests impacted by selective timber harvesting, and reclassified as "secondary"). There are concerns that palm oil buyers will continue to breach their "*No Deforestation*" commitments because of incomplete transparency [44]. Political changes may reverse forest policies with dramatic implications, as seen in Brazil. In response to the covid-19 pandemic the Indonesian government has relaxed or removed several forest regulations. These include removing licenses certifying that the wood comes from legal sources for timber export [45], allowing cleared lands within "protection forest" zones to be converted to so called "food estates" [46]. The Indonesian government also amended a host of other environmental and labour regulations in the so-called "Omnibus Bill" of October 2020, which affects 79 existing laws [47]. More recently, in mid-2021, the REDD + partnership with Norway was unilaterally terminated by Indonesia, casting doubts over commitments to control deforestation [48].

On the positive side, deforestation has declined dramatically over the last half-decade. Steps have been taken to protect forests, including through increased community management [49] and are gaining support and momentum. Transparency has improved, partly because of the growing availability of real-time deforestation monitoring tools [44], and independently verified certification criteria [13]. The slow-down in expansion (now at pre-2004 levels) provides an opportunity for the Indonesian government and concerned stake-holders to work together to improve planning and management of oil palm and other plantations [50]. Nevertheless, the price of palm oil has doubled since the start of the COVID-19 pandemic, and demand is expected to increase [4]. In this context the recent slowdown in plantation expansion is best judged a promising pause rather than as an inevitable and terminal decline. While little is certain, expansion may recover. In the meantime, we must collectively invest in encouraging the good practice and transparency that best serves conservation and future generations as well as local and global development needs.

## Supporting information

**S1 Fig. A sequence of annual cloud-free Landsat composites over an area in Papua province.** These images reveal the annual expansion of an industrial plantation. Imagery displayed in false colors (RGB: Short-wave infrared: band 5; Near infrared: band 6; Red: band 4). Here, forest appears green, while recently cleared areas appear pink.
(TIF)

**S2 Fig. Examples of industrial pulp and paper (Acacia or Eucalyptus) plantations seen by Landsat (in 1:50,000 scale).** Imagery displayed in false colors (RGB: Near infrared; Short-wave infrared; Red). Closed-canopy acacia stands appear red to dark red. Recently harvested stands appear bright cyan. forest is dark brown. (a) Network of riparian forest in an acacia plantation on steep terrain. (b) a plantation on flat surface with rectilinear network of roads

and canals. (c) a plantation on steep terrain with fewer forest corridors. (d) a plantation of flat peat swamps, with acacia stands of varying age, and rectilinear network of canals and roads. (TIF)

**S3 Fig. Composite Radar/Sentinel-2 for year 2019.** Imagery displayed in false color composite (RGB: VV,VH, Red). (a) Closed-canopy (mature) oil palm plantations appear green because of the higher backscatter than other vegetation types in the dual cross-polarization bands (VH). (b) Auriga's oil palm base map (black) missed several plantations (green). (c) The final map used the radar data to capture missed oil palm plantations: industrial oil palm (black); smallholder oil palm (light green); industrial acacia plantations (dark green). (TIF)

**S4 Fig. An area in Riau, Sumatra, where coconut plantations were misclassified as oil palm.** (a) provincial government map of oil palm (black) coconut (yellow), and acacia (green). (b) Auriga oil palm base map. (c) In the final map the areas misclassified as oil palm have been removed. (TIF)

**S5 Fig. High-resolution (<1m) image snapshots detecting the presence of oil palm stands for six reference sites (red squares).** The top images show the distinct planting patterns of three types of industrial plantations: mature (closed-canopy) plantation on flat surfaces, with rectilinear trails (left), mature plantation on undulating surfaces, with contour trails (middle), and partly damaged plantation (right). The bottom images show the distinct planting patterns of smallholder plantations: young (open-canopy) plantation on flat surface (left), mature (closed-canopy) plantation (middle), and damaged plantation on flat surface (right). (TIF)

**S6 Fig. Two reference sites (red squares) not labelled 'oil palm'.** We observed no change in land cover between high-resolution imagery (<1 m) taken before 2019 (left images) and 2019 Sentinel-2 composite (10 m) (right images). The sites were labelled 'other'. (TIF)

**S7 Fig. Two reference sites (red squares) labelled 'oil palm'.** We observed change in land cover between high-resolution imagery (<1 m) taken before 2019 (left images) and 2019 Sentinel-2 composite (10 m) (right images). In the top images, clearing typical of industrial plantations (rectilinear grids) appears in 2019. We labelled this site 'industrial oil palm'. In the bottom images, clearing typical of smallholder plantations appears in 2019, and is adjacent to an existing smallholder oil palm plantations seen on the high-resolution imagery. We labelled the site 'smallholder oil palm'. (TIF)

**S8 Fig. A time series of Normalized Burned Area (NBR) values obtained from Landsat surface reflectance images at a reference site (blue dot).** The visual interpretation of original time-series Landsat images corroborated that the area was converted to industrial oil palm plantation in 2014. (TIF)

**S9 Fig. Contingency table and histogram.** Revealing the correspondence between the year of establishment of industrial oil palm plantations verified with Landsat images and the year of establishment reported on the map for all reference sites labelled 'Industrial oil palm' (N = 612). (TIF)

**S10 Fig. Forest converted to industrial oil palm annually from 2001 to 2016 in Indonesia.**
(a) Based on samples (Austin et al. 2019 [10]). (b) Based on wall-to-wall mapping (this study).
(c) Shows the correspondence between both studies. The line represents the fitted regression
model that goes through the origin (zero intercept).
(PNG)

**S11 Fig. Industrial oil palm expansion from 2001 to 2019 by Indonesian region.** Y-axis represent areas (in 1000-ha, note different scales) of the total area of plantations added each year
between by directly clearing forests (light bars, below), or by using areas already cleared (dark
bars). Black areas on the map represent the total area of industrial oil palm plantations in 2019.
(PNG)

**S12 Fig. Smallholder oil palm expansion from 2001 to 2019 by Indonesian region.** Y-axis
represent areas (in 1000-ha, note different scales) of the total area of plantations added each
year between by directly clearing forests (light bars), or by using areas already cleared (dark
bars). Black areas on the map represent the total area of smallholder oil palm plantations in
2019.
(PNG)

**S1 Table. Error matrix.** Description of sample data as an error matrix of reference sites counts
(see S2 Table for recommended estimated error matrix used to report accuracy results).
(DOCX)

**S2 Table. Error matrix.** Description of sample data as an error matrix of reference sites populated by estimated proportions of area.
(DOCX)

**S3 Table. Share of deforestation caused by oil palm and pulp&paper expansion from 2001
to 2019 for Indonesia and by region.**
(DOCX)

## Acknowledgments

DS's time was covered by NMBU. We thank David B. Clark and two anonymous reviewers for
providing helpful comments on earlier drafts of the manuscript.

## Author Contributions

**Conceptualization:** David L. A. Gaveau.

**Data curation:** Mohammad A. Salim, Husnayaen.

**Formal analysis:** Bruno Locatelli, Arild Angelsen.

**Methodology:** David L. A. Gaveau.

**Resources:** Timer Manurung.

**Validation:** David L. A. Gaveau, Mohammad A. Salim, Adrià Descals.

**Writing – original draft:** David L. A. Gaveau, Bruno Locatelli, Douglas Sheil.

**Writing – review & editing:** David L. A. Gaveau, Arild Angelsen, Erik Meijaard, Douglas
Sheil.

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
