## [Decision Letter · Decision Letter 0]

15 Dec 2021

PONE-D-21-26888

Slowing deforestation in Indonesia follows declining oil palm expansion and lower oil prices

PLOS ONE

Dear Dr. Gaveau,

Thank you for submitting your manuscript to PLOS ONE. After careful consideration, we feel that it has merit but does not fully meet PLOS ONE’s publication criteria as it currently stands. Therefore, we invite you to submit a revised version of the manuscript that addresses the points raised during the review process.

ACADEMIC EDITOR:

Please make more clarifications on the methods used in the paper.

We look forward to receiving your revised manuscript.

Kind regards,

RunGuo Zang

Academic Editor

PLOS ONE

“This work was funded by the WWF-US and Global Environment Facility (GEF) under the Good Growth Partnership, and in collaboration with the Trase Initiative. DS’s time was covered by NMBU.”

 “This work was funded by the WWF-US and Global Environment Facility (GEF) under the Good Growth Partnership, and in collaboration with the Trase Initiative. The funders had no role in study design, data collection and analysis, decision to publish, or preparation of the manuscript”

3. We note that Figure(s) 2 & 3 in your submission contain [map/satellite] images which may be copyrighted. All PLOS content is published under the Creative Commons Attribution License (CC BY 4.0), which means that the manuscript, images, and Supporting Information files will be freely available online, and any third party is permitted to access, download, copy, distribute, and use these materials in any way, even commercially, with proper attribution. For these reasons, we cannot publish previously copyrighted maps or satellite images created using proprietary data, such as Google software (Google Maps, Street View, and Earth). For more information, see our copyright guidelines: http://journals.plos.org/plosone/s/licenses-and-copyright.

a. You may seek permission from the original copyright holder of Figure(s) 2 & 3 to publish the content specifically under the CC BY 4.0 license. 

Additional Editor Comments:

The comments from the referees are postive to your manuscript.What you need in the revision is to make more clarifications on your methods.

Reviewers' comments:

Reviewer's Responses to Questions

**Comments to the Author**

1. Is the manuscript technically sound, and do the data support the conclusions?

Reviewer #1: Yes

Reviewer #2: Partly

Reviewer #3: Yes

2. Has the statistical analysis been performed appropriately and rigorously? 

Reviewer #1: Yes

Reviewer #2: N/A

Reviewer #3: Yes

3. Have the authors made all data underlying the findings in their manuscript fully available?

Reviewer #1: No

Reviewer #2: Yes

Reviewer #3: Yes

4. Is the manuscript presented in an intelligible fashion and written in standard English?

Reviewer #1: Yes

Reviewer #2: Yes

Reviewer #3: Yes

5. Review Comments to the Author

Reviewer #1: General comments:

I initially declined to review this paper, citing my lack of expertise in land use change detection and quantification, as well as having no background in economics. I did however state I was willing to review as a general non-subject-expert tropical forest ecologist, and I accepted the second review invitation under that condition. I advise the Subject Editor to make sure the other reviews are from subject-matter experts, and to consider my review as addressing more basic aspects common to any scientific paper.

As a general scientific reader (see caveats above) I found the paper interesting and readable. It addresses an important topic and is in general well-written and clearly presented.

Data are stated to be available “for interactive viewing” at a website. However it would be impossible for a reader to know exactly what data from that website were used, and there is no guarantee that those data will be available over the long term. I suggest that authors provide the exact data that were used in their analyses, either in a separate data deposition on Dryad or the like, or as Supplemental Information. Websites and their databases change and disappear, whereas depositing the exact data used to produce this paper on Dryad or as SI “freezes” the exact data used in this paper. Just providing a web link for interactive viewing definitely does not comply PLOS One’s data availability policy of providing the “data required to replicate all study findings reported in the article, as well as related metadata and methods”

This manuscript was difficult to review because it did not follow PLOS One formatting guidelines, which require line numbers: “Include page numbers and line numbers in the manuscript file.” Lacking line numbers to refer to, I used several words of the text in brackets instead; the authors will have to use text search to find the section I’m referring to. References in the text also do not follow PLOS One format (brackets instead of superscripts).

Specific comments:

The title seems logically reversed. The assertion is that lower palm oil prices drive slower deforestation and oil palm plantation expansion, so the more logical would seem to be

“Lower palm oil prices are associated with decreased deforestation and oil palm expansion in Indonesia”

The distinction between industrial and smallholder farms should be quantified when it is Introduced in the Introduction. What is the upper bound in ha for small holder, and the lower bound in ha for an industrial plantation?

Putting Materials and Methods after Results is permitted by PLOS One manuscript guidelines. My question to the Authors though is, does that make sense? How could one intelligently read a Results section without first reading the Methods? If one has to read the Methods to understand the Results, why not put the Methods first so the reader can read the paper in logical order?

[Smallholder plantations are smaller] – Give some indication of the size range, for example a mean + SEM in ha.

Table 1. Define “industrial” in “rapid conversion to industrial”

Reviewer #2: Oil palm expansion in the Southeast Asia plays an important role in forest conversion in the region. Spatiotemporal analyses of changes in oil palm distribution and deforestation help quantify this role and contribute to scientifically sound management of the land. There are existing studies and mappings of oil palm expansion in Indonesia, and this manuscript tries to improve such studies by providing time-series mapping of oil palm distribution and separating industrial and smallholder oil palm plantations. The results highlighted declining oil palm expansion and slowing deforestation, which are in general aligned with the existing studies.

1) Time-series mapping of smallholder oil palm plantations is an important contribution of the manuscript; however, the approach is questionable. The authors used an existing map of smallholder oil palm plantations in 2016-2018 and the time series of tree loss maps to derive the annual distributions back to the period of 2001–2019. This method implicitly assumes that all the distributions in the years of 2001 – 2019 are bounded by the existing map in 2016 – 2018. This assumption needs to be justified.

2) The delineation and classification of two types of oil palm plantations are mostly relying on the visual interpretation of satellite images. In such case, survey to local experts is suggested to verify the mapping results.

3) Current ground truthing is mostly based on the checking of 3440 pixels in 2019. Since the manuscript focuses on the time series mapping in 2001 – 2019, the ground truthing pixels/year should be spread to the different time periods such as the beginning, the middle, and the late periods of 2001 – 2019.

4) The Kappa statistic is a more reliable index on classification accuracy, which is missing in the results.

Reviewer #3: This is a superbly executed and clearly written and presented study on a very important topic. The manuscript provides long-overdue clarity on patterns of forest loss and palm oil expansion, including for smallholder palm oil, for the last 20 years. I have only a couple of minor comments for revision. There are obvious challenges in mapping such a complex land use system, especially for smallholders, but the authors are to be commended on employing such a comprehensive approach to validation and clearly presenting both commission and omission errors.

Two small comments

1. You report on Page 13 that “We were more concerned to note that Xu et al. (2020)14 reported a peak in expansion in 2016 but subsequent discussions with those authors suggested an artefact due to using multiple data sources with distinct and sometimes inconsistent properties (Xu et al. pers. comm. to Gaveau 2020).” Given the importance of this contrast and unpublished nature of the reference it would be helpful to include more detail on where the “sometimes inconsistent properties” are evident in the analysis by Xu et al.?

2. Page 17. Worth noting that as of 2021 the payment for results agreement between Norway and Indonesia has been terminated by Indonesia

6. PLOS authors have the option to publish the peer review history of their article (what does this mean?). If published, this will include your full peer review and any attached files.

Reviewer #1: **Yes: **David B. Clark

Reviewer #2: No

Reviewer #3: No

---

## [Author Response · Author response to Decision Letter 0]

1 Mar 2022

We thank the reviewers for their constructive comments. In most cases we have addressed their concerns as suggested. In a few cases we have acknowledged the issue and sought an alternative clarification or to make our intentions clearer. There are few examples where there may be a misunderstanding, but in each case we have reviewed/revised to make our text and views simpler to follow. In each case we have replied point by point. In finalising the manuscript, we have also reviewed it several times for coherence and content and thus made minor changes that benefit the text. We believe the revised article is substantially improved. We have included page numbers and line numbers in the revised manuscript. We answer clarify against each point from the reviewers in the uploaded file: Reply to reviewers

---

## [Editor Report · Decision Letter 1]

16 Mar 2022

Slowing deforestation in Indonesia follows declining oil palm expansion and lower oil prices

PONE-D-21-26888R1

Dear Dr. Gaveau,

We’re pleased to inform you that your manuscript has been judged scientifically suitable for publication and will be formally accepted for publication once it meets all outstanding technical requirements.

Kind regards,

RunGuo Zang

Academic Editor

PLOS ONE

Additional Editor Comments (optional):

accept
---

## [Editor Report · Acceptance letter]

21 Mar 2022

PONE-D-21-26888R1 

Slowing deforestation in Indonesia follows declining oil palm expansion and lower oil prices 

Dear Dr. Gaveau:

I'm pleased to inform you that your manuscript has been deemed suitable for publication in PLOS ONE. Congratulations! Your manuscript is now with our production department. 

Kind regards, 

on behalf of

Professor RunGuo Zang 

Academic Editor

PLOS ONE